# GM-Improved Antiaging Effect of Acrylonitrile Butadiene Styrene in Different Thermal Environments

**DOI:** 10.3390/polym12010046

**Published:** 2019-12-28

**Authors:** Yuchao Wang, Ming Chen, Miaoyu Lan, Zhi Li, Shulai Lu, Guangfeng Wu

**Affiliations:** 1ABS Technology Center of PetroChina, Jilin 132021, China; jh_wangyuc@petrochina.com.cn (Y.W.); jh_chming@petrochina.com.cn (M.C.); jh_lanmy@petrochina.com.cn (M.L.); jh_lzhi@petrochina.com.cn (Z.L.); 2School of Chemical Engineering, Changchun University of Technology, Changchun 130012, China

**Keywords:** polymer degradation, acrylonitrile butadiene styrene, convection oven, extrusion, recycling, mechanical properties

## Abstract

A stabilizer called 2-tert-butyl-6-(3-tert-butyl-2-hydroxy-5-methylbenzyl)-4-methylphenyl acrylate (GM) was mixed in acrylonitrile butadiene styrene (ABS) with the same amount of 9-bis(octadecyloxy)-2,4,8,10-tetraoxa-3,9-diphosphaspiro[5.5]undecane (DSPDP), octadecyl-3-(3,5-di-tert-butyl-4-hydroxyphenyl) propionate (Irganox 1076) and tris(3,5-di-tert-butyl-4-hydroxybenzyl) isocyanurate (Irganox 3114) to investigate the influence of additives on the antiaging effect of ABS in oven aging or repeated extrusion aging. It was found that the ABS doped with the GM stabilizer showed a better yellowing resistance and thermal stability than the ABS doped with other antioxidants. Owing to the fact that the stabilizer can act on the free radicals before it has been peroxidized, it could trap the free radicals as a consequence of directly blocking the oxidation process of the active species, thus solving the problem of oxidative degradation of the materials from the source. This work provides guidance for improving thermal stability of ABS, indicating a promising potential for industrial application.

## 1. Introduction

Acrylonitrile butadiene styrene (ABS) resins have attracted considerable attention due to their superior mechanical properties [1], heat resistance [2], chemical resistance [3], recyclability [4,5,6] and ease of processing [7,8], and are widely used in telecommunications, automotive industry, consumer markets and business machines [9,10]. However, ABS is easily excited by the heat [11,12], light [13,14,15], oxygen [16,17] and other conditions during its processing [18,19], leading to aging degradations of the material. Because the presence of unstable carbon–carbon double bonds in the butadiene microstructure generates macromolecular-active radicals during processing, thermal degradation is promoted and results in a rapid yellowing and brittleness of ABS [20,21].

To solve this problem, most of previous reports employed doping strategy to prepare highly aging-resistant ABS resins. Some people chose traditional additives to improve heat resistance. For example, Fiorio’s group found that the addition of primary antioxidants in ABS increases the thermal–oxidative stability, whereas the secondary antioxidants only increase the oxidation peak temperature [22]. Others have chosen new types of additives. Ong’s group showed that pretreatments on filled oil palm fiber can lower the onset thermal degradation temperature of acrylonitrile butadiene styrene composites in differential thermogravimetric analysis [23]. Teh’s group successfully improved the thermal stability of the ABS resin by adding carbon black (CB) to the ABS [24]. Yan’s group enhanced the electrical conductivity and thermal conductivity of the ABS via the addition of reduced graphene oxide (rGO) in the ABS resin, because it can tightly wrap the rGO sheet on the surface of ABS microspheres [25]. In addition, Allen’s group used other stabilizers to conduct a comprehensive study of K-resin by both thermal-aging degradation and photo-aging degradation [26]. However, the doping strategy also brings about changes in the color-phase morphology or other properties thereof. This dilemma makes it difficult to explore the improved thermal stability for the ABS resin [27]. 

Here, we successfully obtained high thermal stability ABS through a doping strategy without affecting the properties of ABS itself. A systematic study was carried out to investigate effects of 2-tert-butyl-6-(3-tert-butyl-2-hydroxy-5-methylbenzyl)-4-methylphenyl acrylate (GM), 9-bis(octadecyloxy)-2,4,8,10-tetraoxa-3,9-diphosphaspiro[5.5]undecane (DSPDP), octadecyl-3-(3,5-di-tert-butyl-4-hydroxyphenyl) propionate (Irganox 1076) and tris(3,5-di-tert-butyl-4-hydroxybenzyl) isocyanurate (Irganox 3114) antioxidants on the thermal aging of ABS in two different thermal environments, a 100 °C convection oven and a 240 °C extruder simulating the extreme environment. It was found that the ABS doped with the GM stabilizer showed better yellowing resistance and thermal stability than the ABS doped with other antioxidants. Owing to the fact that the stabilizer can act on the free radicals before it has been peroxidized, it could trap the free radicals as a consequence of directly blocking the oxidation process of the active species, thus solving the problem of oxidative degradation of the materials from the source. Hence, the GM antioxidant can greatly improve thermal stability of the ABS in the convection oven or the extruder. This work provides guidance for improving thermal stability of ABS, indicating promising potential for industrial application.

## 2. Materials and Methods 

### 2.1. Materials

The materials used in this study were as follows: ABS-grafted powder PW-151 (KunLun, Jilin, China, obtained by emulsion polymerization), SAN resin 2437 (KunLun, Jilin, China), acrylic acid (Sigma Aldrich, Shanghai, China), phosphoryl trichloride (Sigma Aldrich, Shanghai, China), triethylamine (TEAE), 2,2’-methylenebis(6-tert-butyl-4-methylphenol) (Antioxidant 2246, Sigma Aldrich, Shanghai, China), GM (laboratory homemade), DSPDP (Cytec, Jilin, China, phosphite secondary antioxidant), Irganox 1076 (BASF, Shanghai, China, phenolic primary antioxidant) and Irganox 3114 (BASF, Shanghai, China, antioxidant-stabilizer). 

### 2.2. Synthesis of Stabilizer GM

Firstly, 120 g of Antioxidant 2246, 20 g of acrylic acid, 2.0 g of triethylamine and an appropriate amount of the organic solvent were sequentially added to a 1000 mL four-necked flask. After heating in an oil bath under stirring, the temperature was raised to about 110 °C and 30 g of phosphoryl trichloride were added dropwise; the dropwise addition was completed in 45 min. The dropped raw material reacted to release a small amount of hydrogen chloride gas which was absorbed by the hydrogen chloride absorber. The temperature was maintained for 3 h before the reaction was completed. Thereafter, the organic solvent recrystallization method was used to purify the crude reaction product. Finally, a yellowish crystalline product was obtained which is referred to as Stabilizer GM. We use the synthetic method in the article by Yachigo’s group to obtain the uniform-quality stabilizer GM from the raw materials, so as to obtain reliable and repeatable experimental results [28].

### 2.3. Sample Preparation

The ABS grafted powder and the SAN resin were mixed at a mass ratio of 25:75, and four equal parts of the GM, DSPDP, Irganox 1076 and Irganox 3114 were added in in the mixture, then sheared and granulated the mixture by a twin-screw extruder at 240 °C. The granulated ABS-material was subjected to a series of eight cycles of oven aging at 100 °C for 72 h per cycle, or five cycles of anaerobic repeated extrusion at 240 °C. We chose the temperature at 100 and 240 °C was because these temperatures presented greater challenge for the additives to maintain the same performance of the materials. In addition, it was more intuitively to compare the differences of each additives in the protective ability [6,29,30,31,32].

### 2.4. Analysis

#### 2.4.1. Fourier Transform Infrared Spectroscopy (FTIR)

Fourier transform infrared spectroscopy (FTIR, IR Prestige-21, Shimadz, Kyoto, Japan) was used to evaluate the four additives and changes in ABS samples through accelerated degradation. All samples were studied in attenuated total reflectance (ATR) mode from 4000 to 600 cm^−1^. The spectra of ABS samples form an absorbance peak of carbonyl at 1720 cm^−1^ during degradation, which was compared to the absorbance peak of acrylonitrile at 2237 cm^−1^. The integral ratio between the absorbance peak at 1720 cm^−1^ (carbonyl) and that at 2237 cm^−1^ (acrylonitrile) was defined as the carbonyl index, the equation is as follows [33,34,35]: (1)$$Carbonyl\;index=\frac{\mathrm{Transmittance}\;\mathrm{at}\;1720{\;\mathrm{cm}}^{-1}}{\mathrm{Transmittance}\;\mathrm{at}\;2237{\;\mathrm{cm}}^{-1}}\;.$$

#### 2.4.2. Thermogravimetric Analysis (TGA)

All raw materials were characterized simultaneously by thermogravimetric analysis (STA 449 F5, Netzsch, Selb, Germany). Samples (for the results, see Appendix A) of 10 ± 2 mg were heated from 25 to 600 °C at a heating rate of 10 °C min^−1^ in a Pt–Rh pan [36]. The shielding gas was high-purity nitrogen and the flow rate set to 20 mL·min^−1^. The purge gas was air and the flow rate set to 20 mL·min^−1^.

#### 2.4.3. Color Measurements 

The color properties of the samples were investigated with a spectrophotometer (UltraScan PRO, Hunterlab, Reston, VA, USA) using a Artificial Daylight 6500K (D65) light source and 10° viewing angle. All color measurements were obtained from the compression molded films.

The color difference (Δ*E*) between the two sets of injection samples was tested. One was that the temperature of the cutting section of the plastic cylinder was 220 °C and the residence time of the cylinder 40 s. The other was that the temperature of the cutting section of the plastic cylinder was 250 °C and the residence time 180 s. The difference between the heat resistance of the ABS resin additives was characterized by testing the color difference values of ABS samples under these two conditions.

The results show three colorimetric coordinates: luminosity (*L*), red/green component (*a*) and blue/yellow component (*b*). The total color difference (Δ*E*) between the samples caused by temperature or time change was calculated using Equations (2) and (3) [35]: (2)$$\Delta E=\sqrt{\Delta L^{2}+\Delta a^{2}+\Delta b^{2}}$$
(3)$$\Delta L=L_{250℃}-L_{220℃},\;\Delta a=a_{250℃}-a_{220℃},\;\Delta b=b_{250℃}-b_{220℃},$$
where Δ*L* is the difference of the injection-molded-samples between 250 °C and 220 °C, Δ*a* is red/green component difference between the samples under the two different temperatures and Δ*b* is the difference of the blue/yellow component.

The yellowing index (YI) was obtained by the International Commission on illumination (CIE) tristimulus values *X*, *Y* and *Z* according to the CIELAB color system [37]. The YI was determined according to Equation (4) [22]:(4)$$\mathrm{YI}=\frac{100\times \left(1.3013X-1.1498Z\right)}{Y}.$$

#### 2.4.4. Mechanical Testing

Notched-impact strength and melt-index rate (MFR) were measured according to ASTM-D256 and ASTM-D1238 standards, respectively.

## 3. Results

### 3.1. Additive Selection

To systematically study and understand the influence of additives on the antiaging effect of ABS, the additives GM, DSPDP, Irganox 1076 and Irganox 3114 were selected to dope the ABS. The molecular structure of the additives is shown in Figure 1a. Figure 1b shows the Fourier transform infrared spectroscopy of the additives. In our experiment, we synthesized the stabilizer GM. As shown in Figure 1b, 804 and 856 cm^−1^ are the bending vibration peaks of the benzene ring; the stretching vibration peak of the benzene appears at 1599 cm^−1^, which proves that the synthesized GM material does contain the benzene ring group. The –OH bending vibration peak appears at 1358 cm^−1^, indicating that the synthesized GM material contains the phenolic hydroxyl functional group. In addition, the 1161, 1633 and 1729 cm^−1^ peaks are C–O, C=C and C=O stretching vibration peaks, respectively, indicating that the microscopic benzene ring structure of the sample is attached with an acrylic group. In summary, the prepared material is a desired GM product containing a bisphenol acryloyl group. Figure 1b also shows the Fourier transform infrared spectroscopy of DSPDP, Irganox 1076 and Irganox 3114, three other additives, showing POC, Ar–OH and =CH, three characteristic functional groups appearing at 1032, 3638 and 852 cm^−1^.

The four kinds of additives we selected play different roles in the thermal-oxidation process. The GM stabilization mechanism is confirmed to be a unique bifunctional mechanism, which consists of polymer radical trapping by the acrylate group, followed by rapid hydrogen transfer from intramolecular hydrogen-bonded phenolic hydroxyl groups to form stable phenoxyl radicals. Therefore, the problem of oxidative degradation of materials is solved from the source of the thermal-oxidation process (for the schematic, see Appendix A). DSPDP acts as a hydroperoxide scavenger that rapidly decomposes macromolecular hydrogen peroxide, stopping the thermal-oxidation process and thus preventing continued aging. Irganox 1076 and Irganox 3114 act as radical scavengers that react with macromolecular peroxidative free radicals to inhibit the partial thermal-oxidation process and hence slows aging. It can be confirmed that GM has a unique heat-resistant effect compared with traditional primary and secondary antioxidants.

### 3.2. Carbonyl Index

To explore the influence of additives on the antiaging effect of ABS, we calculated the carbonyl index (CI) of ABS with different additives. The carbonyl index is defined as the integral ratio because the absorbance peaks of acryloacrylonitrile are usually unchanged during degradation progress. The oxidation of the polybutadiene rubber phase of ABS material would form carbonyl groups in the degradation, so that the degradation of the material can be evaluated from the change of the carbonyl absorption peak. The Fourier transform infrared spectroscopy of ABS with different additives after eight cycles in the oven aging at 100 °C for 72 h is shown in Figure 2a. Figure 2b shows the dependence of carbonyl index for ABS with different additives on the number of cycles in the oven aging at 100 °C for 72 h. It can be clearly observed that the carbonyl index value of ABS with the GM is the lowest, only 0.68, which is much smaller than that of ABS with DSPDP, Irganox 1076 and Irganox 3114. This result indicated that the ABS with the GM exhibited the least aging, and the additive of GM successfully improved the antiaging effect of ABS. It is confirmed that the effect of the GM additive, which solves the problem of oxidative degradation of materials from the source of the thermal-oxidation process, and is stronger than that of the additive of DSPDP (hydroperoxide scavenger), Irganox 1076 or Irganox 3114 (radical scavengers).

### 3.3. Thermal Properties

To further explore the influence of additives on the antiaging effect of ABS, we evaluated the temperature of ABS with different additives under different weight loss in the two different thermal environments. Figure 3 shows the temperature from thermogravimetric analysis (TGA) of different weight loss for ABS with different additives after eight cycles in the oven aging at 100 °C for 72 h. The initial 1% decomposition temperature (*T*_1%_) of ABS with different additives in the convection oven is shown in Figure 3a. It can be observed that the initial decomposition temperatures of ABS with GM, DSPDP, Irganox 1076 and Irganox 3114 all show a tendency to decrease first and then increase as the number of aging cycles increased. The temperature drop is due to the partial aging of the added chemicals in the aerobic environment during the first three cycles. When the sample was placed in the thermogravimetric analyzer with similar high temperature, the added group was oxidized prior to the ABS sample, which lowered the overall initial *T*_1%_ decomposition temperature. The *T*_1%_ rose after 3–5 cycles because the addition of antioxidant was almost oxidized and could no longer affect the decomposition temperature of the ABS resin. Moreover, the 1% mass decomposition temperature of ABS with GM and DSPDP is almost higher than that of ABS with Irganox 1076 and Irganox 3114. This phenomenon also appears for *T*_3%_ and *T*_5%_, shown in Figure 3b,c, which indicates that the acrylate functional group, unique to GM, and the phosphite structure contained in DSPDP play an important role in resistance to the thermal–oxidative degradation of polymers. 

In addition, dependence of the initial temperature on the weight loss for ABS with different additives after extrusion steps is shown in Figure 4. During the extrusion process, it is easy to produce a high temperature and anoxic environment when the material is filled between the screw and the barrel. Figure 4a shows the initial degradation temperature of ABS with different additives after one extrusion process. It can be seen that the thermal decomposition temperature of ABS with GM, DSPDP, Irganox 1076 and Irganox 3114 are approximately the same, and as the percentage of weight loss increased, the trend of thermal decomposition temperature remained consistent. As the number of cycles increased to three (Figure 4b), the thermal decomposition temperature of ABS with GM was higher than that of ABS with other additives. This is because the acrylate structure of bisphenol monoacrylate can quickly capture macromolecule free radicals at the initial stage (Equation (5)), thus preventing macromolecules from self-crosslinking (Equation (6)). The special bisphenol monoacrylate generates a relatively stable radical ending reaction. As shown in Figure 4c,d, the stabilizer GM can similarly increase the heat resistance of the ABS sample after 5–6 extrusions.
(5)R• + ln H →RH+ln•
(6)2R• →R–R
where R• represents macromolecule free radicals, RH represents stable macromolecule.

### 3.4. Optical and Mechanical Properties

Last but not least, we also explored the influence of additives on the optical and mechanical properties of ABS in two different thermal environments. Figure 5 and Figure 6 show the optical and mechanical properties of ABS with different additives after aging within the convection oven and the extruder. The additive of GM not only improved the thermal stability of ABS, but it also had a good color promotion in the two different thermal environments. As shown in Figure 5a and Figure 6a, it can be observed that the ABS with different additives yellowed after oven aging or repeated extrusion aging. The yellow index of ABS with different additives increased as well as the color difference given in Figure 5b and Figure 6b.

According to the Equation (7), and further combining the yellow index with the color difference index, the phosphite group contained in the ABS with DSPDP reacts faster to decompose hydroperoxide when compared to other samples:(7)$$\mathrm{ROOH}\;+\;2(R’O)3\mathrm{POH}\;\to 2{\left(R’O\right)}_{3}P=O+\mathrm{ROH}+H_{2}O$$
where R repersents the alkyl group in macromolecular, R’ repersents the alkyl group in DSPDP.

The increase of color stability of ABS with Irganox 1076 indicates that the hindered phenolic antioxidant easily forms anthraquinone after absorbing the peroxy radical, causing the material to turn yellow.

Furthermore, a cantilever-beam impact tester was used to analyze the notched impact strength (IZOD) of ABS with different additives after aging by the convection oven or the extruder. In addition, the molecular mobility of ABS with different additives was analyzed using a melt-flow tester. The notched impact strength of ABS with different additives is shown in Figure 5c and Figure 6c, and the melt flow rate (MFR) is shown in Figure 5d and Figure 6d. As shown in Figure 5c,d, the impact toughness and molecular fluidity of ABS with different additives did not significantly attenuate when compared to pure ABS (for the results, see Appendix A) after aging by the convection oven, indicating that the resin itself had no serious cross-linking phenomenon, and the addition of the antioxidant had no effect on the impact strength and melt flow rate. Figure 6c,d demonstrate that the ABS undergoes considerable cross-linking, which shows an evident decrease in melt flow rate. The ABS sample with GM increased its impact strength and melt flow rate of ABS, indicating that the bifunctional group in GM can prevent the macromolecular radical cross-linking curing caused by less oxygen in repeated extrusion.

## 4. Conclusions

In summary, we successfully obtained high thermal stability and a better yellowing-resistant ABS with the GM additive in the convection oven or the extruder. In the oxygen convection oven environment, the polybutadiene double bond of the ABS was attacked by the active free radical to form a carbonyl group. It can be clearly observed from the FTIR that the carbonyl index value of ABS with the GM is the lowest, which is much smaller than that of ABS with DSPDP, Irganox 1076 and Irganox 3114. As the number of aging cycles increased, the thermal decomposition temperature (TGA) of ABS with GM was higher than that of ABS with other additives. These results indicated that the difunctional structure of GM could reduce the formation of carbonyl group by the reaction with free radicals prior to aging degradation. It also proved that the acrylate structure of bisphenol monoacrylate can rapidly capture macromolecular free radicals in the initial stage, generate relatively stable free radicals and terminate the reaction. In the extrusion environment with less oxygen, the lower yellow index and smaller chromatic aberration demonstrated better color retention of ABS sample with GM. The higher melt flow rate of ABS sample with GM also indicated its excellent ability to capture macromolecular radicals. It could effectively inhibit the cross-linking of polymers, slow the aging of materials and improve the heat-aging resistance of ABS.

## Figures and Tables

**Figure 1 polymers-12-00046-f001:**
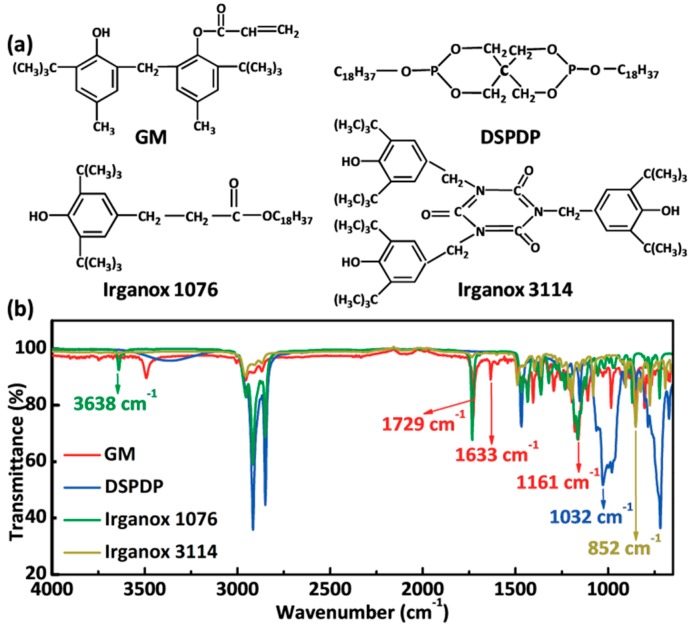
(**a**) Molecular structure of different additives; 2-tert-butyl-6-(3-tert-butyl-2-hydroxy-5-methylbenzyl)-4-methylphenyl acrylate (GM), 9-bis(octadecyloxy)-2,4,8,10-tetraoxa-3,9-diphosphaspiro[5.5]undecane (DSPDP), octadecyl-3-(3,5-di-tert-butyl-4-hydroxyphenyl) propionate (Irganox 1076) and tris(3,5-di-tert-butyl-4-hydroxybenzyl) isocyanurate (Irganox 3114). (**b**) Fourier transform infrared spectroscopy of the additives.

**Figure 2 polymers-12-00046-f002:**
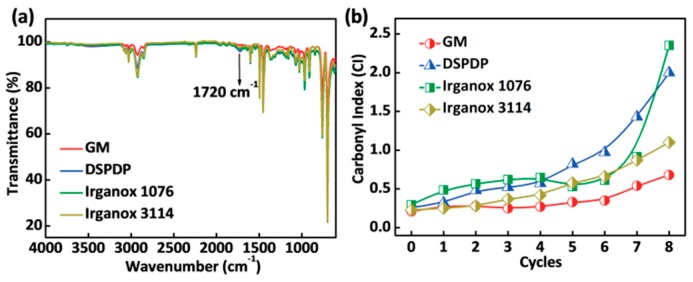
(**a**) Fourier transform infrared spectroscopy (FTIR) of acrylonitrile butadiene styrene (ABS) with different additives after eight cycles at 100 °C. (**b**) Carbonyl index versus aging cycles for ABS with different additives at 100 °C.

**Figure 3 polymers-12-00046-f003:**
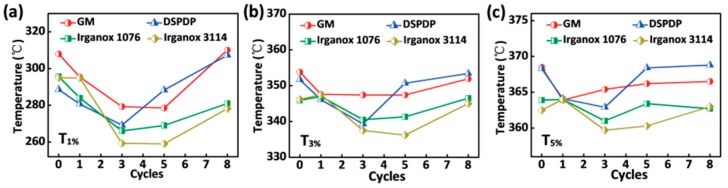
(**a**) Temperature of 1% weight loss for ABS with different additives; (**b**) temperature of 3% weight loss for ABS with different additives; and (**c**) temperature of 5% weight loss for ABS with different additives.

**Figure 4 polymers-12-00046-f004:**
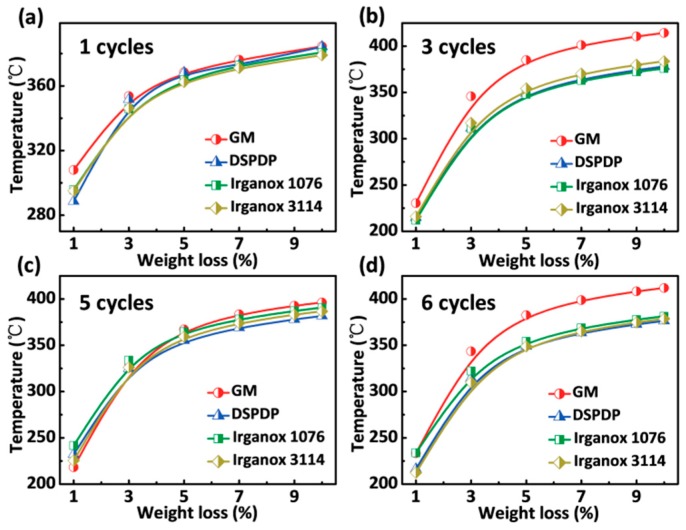
Dependence of the initial temperature on the weight loss for ABS with different additives after different extrusion times: (**a**) the first extrusion; (**b**) the third extrusion; (**c**) the fifth extrusion; and (**d**) the sixth extrusion.

**Figure 5 polymers-12-00046-f005:**
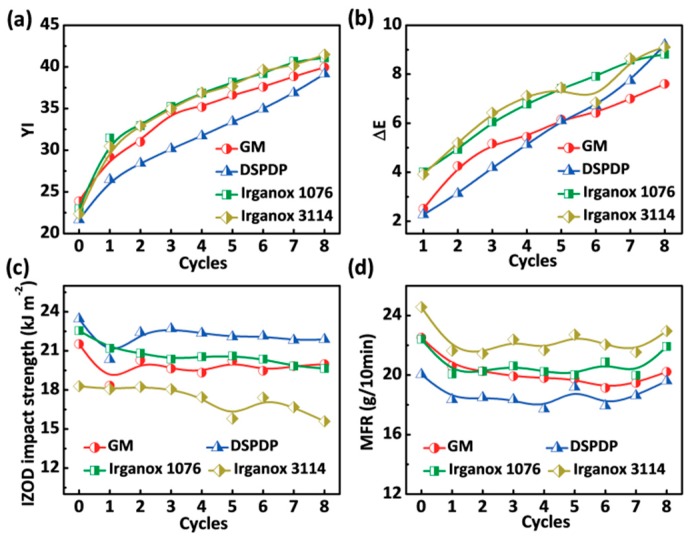
Optical and mechanical properties versus aging cycles for ABS: (**a**) yellow index (YI); (**b**) color difference value (Δ*E*); (**c**) notched impact strength; and (**d**) melt flow rate (MFR).

**Figure 6 polymers-12-00046-f006:**
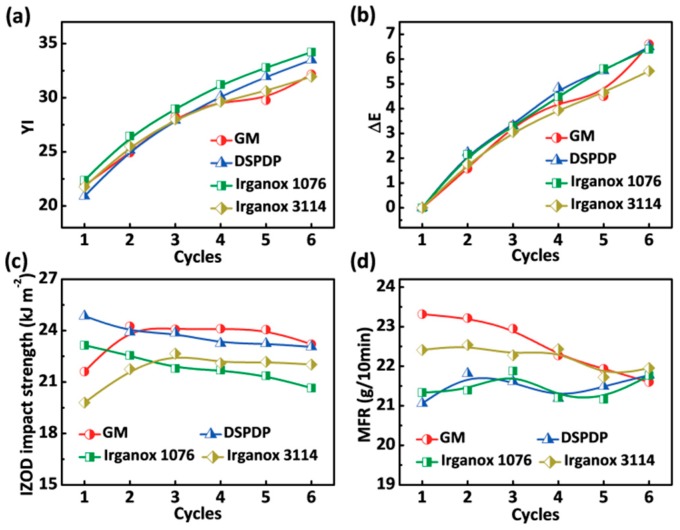
Optical and mechanical properties versus repeated extrusion for ABS: (**a**) yellow index (YI); (**b**) color difference value (Δ*E*); (**c**) notched impact strength; and (**d**) melt flow rate (MFR).

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
