# Peer review of "GM-Improved Antiaging Effect of Acrylonitrile Butadiene Styrene in Different Thermal Environments"

_polymers, 2019, doi:10.3390/polym12010046_

Round 1

Reviewer 1 Report

Line 83-84: Why are they chosen as aging temperatures 100 and 240 ?.

In the legend of Figure s1, the temperature units do not appear. In periods of degradation with resins it would be advisable to reach the degradation temperature of 900 ºC, to avoid the appearance of some degradation process at high temperatures. What is the reason for carry out experiments only up to 600ºC?

Equations 3 and 4: Although they are known equations should cite some reference. 

Line 176: The title of this section is repeated on line 222.

The legends of the figures are too long, authors should shorten them.

The conclusions should reflect the results of TGA and FT-IR.

Author Response

Manuscript ID: polymers-657420

Manuscript title: GM-improved anti-aging effect of acrylonitrile-butadiene-styrene in different thermal environments

Reviewer 1

Our response: Thank you very much for your valuable and insightful comments for our manuscript. We highly appreciate your positive comments and detailed suggestions on our manuscript. Your suggestions give us deeper consideration on our manuscript. The following revisions were made on the basis of your comments. All revisions have been highlighted by yellow color in our revised manuscript.

Comment 1

Reviewer wrote: Line 83-84: Why are they chosen as aging temperatures 100 °C and 240 °C?

Our response: Thank you very much for your important and valuable comments for our manuscript. According to the previously reported article, it is well known that the physical properties, especially elongation and impact strength of the ABS material can age in an oven at 50~90 °C and show conspicuous deterioration. [Journal of Applied Polymer Science 1968, 12, 655; Polymer Engineering & Science, 1976, 16, 265; Polymer degradation and stability; 2003, 81, 359; Journal of thermal analysis, 1996, 46, 465]. A service time of about 1 year at room temperature can be simulated by an ageing time of about 72 hours at 90 °C with regard to thermo-oxidative degradation. This state can be reached in less time at 100 °C. Therefore, according to the foregoing content, the purpose of selecting 100 °C in this article instead of 50~90 °C is to increase the aging experiment temperature, so the aging phenomenon is more likely to occur, and it is more intuitively to show both advantages and disadvantages of different types of antioxidants. In addition, many previously reports have reported that the temperature range of ABS material during extrusion is generally 200~235 °C. The reason we chose the temperature at 240 °C is that increasing the temperature makes the materials more prone to aging, so the challenge for the additives to maintain the performance of the materials is greater, and the differences in the protective ability of each additive can be compared more intuitively. Hence, we chose 100 °C and 240 °C as the aging temperature in the oven and extrusion process, respectively. According to your suggestion, we modified our manuscript.

Our revision:

Line 89: add “The reason we chose the temperature at 100 and 240 °C is that the challenge for the additives to maintain the performance of the materials, and the differences in the protective ability of each additive can be compared more intuitively.”

Notes and references: add

[29] Shimada, J.; Kabuki, K. The mechanism of oxidative degradation of ABS resin. Part I. The mechanism of thermooxidative degradation. Journal of Applied Polymer Science 1968, 12, 655-669.

[30] Wyzgoski, M. G. Effects of oven aging on ABS, poly (acrylonitrile‐butadiene‐styrene). Polymer Engineering & Science 1976, 16, 265-269.

[31] Boldizar, A.; Möller, K. Degradation of ABS during repeated processing and accelerated ageing. Polymer degradation and stability 2003, 81, 359-366.

[32] Rosik, L.; Kovářová, J.; Pospíšil, J. Lifetime prediction of ABS polymers based on thermoanalytical data. Journal of thermal analysis 1996, 46, 465-470.

[33] Scaffaro, R.; Botta, L.; Di, B. G. Physical properties of virgin-recycled ABS blends: Effect of post-consumer content and of reprocessing cycles. European Polymer Journal 2012, 48, 637-648.

Comment 2

Reviewer wrote: In the legend of Figure s1, the temperature units do not appear. In periods of degradation with resins it would be advisable to reach the degradation temperature of 900 °C, to avoid the appearance of some degradation process at high temperatures. What is the reason for carry out experiments only up to 600 °C?

Our response: Thank you very much for your important and valuable comments for our manuscript. Previous studies on the degradation of ABS resins have shown that the evolution of butadiene begins at 290 °C and it continues to appear throughout the run. Small absorptions in the aromatic C-H region appear near 400 °C, and their evolution has ceased by 500 °C. A residue of 9% remains at 600 °C of ABS. When the temperature continues to rise thereafter, the sample mass does not change much. [Polymer Degradation and Stability, 1995, 47, 217] Therefore, we carried out experiments only up to 600 °C.

Our revision:

Notes and references: add

[37] Suzuki, M.; Wilkie, C. A. The thermal degradation of acrylonitrile-butadiene-styrene terpolymei as studied by TGA/FTIR. Polymer Degradation and Stability 1995, 47, 217-221.

Comment 3

Reviewer wrote: Equations 3 and 4: Although they are known equations should cite some reference. 

Our response: We really appreciate your valuable question. According to your suggestion, we have modified our manuscript to make it clear to more readers.

Our revision:

Notes and references: add

[38] Santos, R. M., Botelho, G. L., Machado, A. V. Artificial and natural weathering of ABS. Journal of applied polymer science 2010, 116, 2005-2014.

Comment 4

Reviewer wrote: Line 176: The title of this section is repeated on line 222.

Our response: Thank you for your insight and remind us of our mistakes. According to your suggestion, we have modified our manuscript to avoid more readers misunderstanding.

Our revision:

Line 224: change to "Optical and mechanical properties”.

Comment 5

Reviewer wrote: The legends of the figures are too long, authors should shorten them.

Our response: Thanks very much for your suggestions on our manuscript. According to your suggestion, we have modified our manuscript.

Our revision:

Line 180: change to "Figure 2. (a) FT-IR of ABS with different additives after 8 cycles at 100℃; (b) Carbonyl index versus aging cycles for ABS with different additives at 100℃."

Line 201: change to "Figure 3. (a) Temperature of 1% weight loss for ABS with different additives; (b) Temperature of 3% weight loss for ABS with different additives; (c) Temperature of 5% weight loss for ABS with different additives."

Line 221: change to “Figure 4. Dependence of the initial temperature on the weight loss for ABS with different additives after different extrusion times: (a) the first extrusion; (b) the third extrusion; (c) the fifth extrusion; (d) the sixth extrusion.”

Line 254: change to “Figure 5. Optical and mechanical properties versus aging cycles for ABS: (a) Yellow index (YI); (b) Color difference value (ΔE); (c) Notched impact strength; (d) Melt flow rate (MFR).”

Line 257: change to “Figure 6. Optical and mechanical properties versus repeated extrusion for ABS: (a) Yellow index (YI); (b) Color difference value (ΔE); (c) Notched impact strength; (d) Melt flow rate (MFR).”

Supporting Information: change to “Figure S1. TGA results for ABS with different additives after different extrusion times: (a) the first extrusion; (b) the third extrusion; (c) the fifth extrusion; (d) the sixth extrusion.”

Comment 6

Reviewer wrote: The conclusions should reflect the results of TGA and FT-IR.

Our response: Thank you very much for your important and valuable comments for our manuscript. According to your suggestion, we reflect the results of TGA and FT-IR in the conclusion section.

Our revision:

Line 265: add "It can be clearly observed from the FT-IR that the carbonyl index value of ABS with the GM is the lowest, which is much smaller than that of ABS with DSPDP, Irganox 1076 and Irganox 3114. As the number of aging cycles increased, the thermal decomposition temperature (TGA) of ABS with GM was higher than that of ABS with other additives. These results indicated that the difunctional structure of GM could reduce the formation of carbonyl group by the reaction with free radicals prior to aging degradation. And it is proved that the acrylate structure of bisphenol monoacrylate can rapidly capture macromolecular free radicals in the initial stage, generate relatively stable free radicals and terminate the reaction."

Reviewer 2 Report

This paper cannot be published in its present form. The authors have not acknowledged that the stabilizer "GM" that they have synthesized was already commercially available from Sumitomo Chemical Corp from the 1990's as Sumilizer GM. There is no mention of the references to the comprehensive research published in Polymer Degradation and Stability from 1988 onwards by Yachigo et. al. on the role these additives may play in stabilizing butadiene-containing polymers (such as ABS).

The novelty of what the authors have done compared to these and other authors needs to be explained, otherwise the paper lacks sufficient novelty to warrant publication.

Regarding the role of the other stabilizers in the formulations, the authors also do not mention the comprehensive study by Allen et. al. on coloration of the closely related SBC. (Polymer Degradation and Stability, 2006, 91:1395.).

This paper is marginal for publication in any case as the methods used are very routine and there is no real mechanistic contribution to the literature on polymer degradation and stabilization. 

Author Response

Manuscript ID: polymers-657420

Manuscript title: GM-improved anti-aging effect of acrylonitrile-butadiene-styrene in different thermal environments

Reviewer 2

Our response: Thank you very much for your valuable and professional suggestions for our manuscript. We really appreciate your patient comments on our manuscript. Your comments guided us to think our results more deeply. We believe your suggestions could improve our present work greatly. Here we can answer your comments one by one. All revisions have been highlighted by yellow color in our revised manuscript.

Comment 1

Reviewer wrote: This paper cannot be published in its present form. The authors have not acknowledged that the stabilizer "GM" that they have synthesized was already commercially available from Sumitomo Chemical Corp from the 1990's as Sumilizer GM. There is no mention of the references to the comprehensive research published in Polymer Degradation and Stability from 1988 onwards by Yachigo et. al. on the role these additives may play in stabilizing butadiene-containing polymers (such as ABS).

Our response: Thank you very much for your valuable and professional suggestions for our manuscript. We really appreciate your patient comments on our manuscript. We apologize to you for misunderstanding our article because of our unclear expression. Therefore, we explain as follows: (1) We know that the stabilizer "GM" was already commercially available from Sumitomo Chemical Corp from the 1990's as Sumilizer GM.[Polymer Degradation and Stability, 1988,22,63] Although GM has been commercialized in Japan's Sumitomo company for 30 years, the quality of several kinds of commercialized GM I purchased is different, making it impossible for us to obtain repeated experimental results. Therefore, we use the synthetic method in the literature to obtain the uniform quality stabilizer "GM" from the raw materials, so as to obtain reliable and repeatable experimental results. (2) The comprehensive research published in Polymer Degradation and Stability from 1988 onwards by Yachigo et. al. explained that the stabilizer GM shows the unique thermal stabilizing effect in butadiene-containing polymers. As the reviewer said, the stabilizer GM may play a role in stabilizing ABS. However, in our manuscript, different from Yachigo's application of GM in SBS materials for experiments, our focus is to compare the anti-aging effects of GM, DSPDP, Irganox 1076 and Irganox 3114 on ABS in different thermal environments, such as in an oven or repeated extrusion. We apologize again for the misunderstanding. According to your suggestion, we revised our manuscript to make it clear to more readers.

Our revision:

Line 11: delete "synthesized and."

Line 81: add “We use the synthetic method in the article by Yachigo’s group to obtain the uniform quality stabilizer GM from the raw materials, so as to obtain reliable and repeatable experimental results [38].”

Notes and references: add

[28] Yachigo, S.; Sasaki, M.; Takahashi, Y.; Kojima, F.; Takada, T.; Okita, T. Studies on polymer stabilisers: Part I—A novel thermal stabiliser for butadiene polymers. Polymer Degradation and Stability 1988,22,63-77.

Comment 2

Reviewer wrote: The novelty of what the authors have done compared to these and other authors needs to be explained, otherwise the paper lacks sufficient novelty to warrant publication.

Our response: Thanks very much for your suggestions. As mentioned before, Yachigo's article focuses on whether GM is effective in SBS materials. However, our manuscript focuses on exploring whether the performance of ABS materials has changed after aging with GM. In addition, previous papers on ABS often discussed the effect of antioxidants on the thermal aging degradation of ABS materials and the synergistic effect of primary and secondary antioxidants. Our manuscript focuses on the difference of GM from traditional primary and secondary antioxidants, it has unique heat-resistant effect, and this conclusion has been proved by aging simulation experiments. According to your suggestion, to highlight the novelty of our work, we revised our manuscript to make it clear to more readers.

Our revision:

Line 161: add "It can be confirmed that GM has unique heat-resistant effect compared with traditional primary and secondary antioxidants."

Comment 3

Reviewer wrote: Regarding the role of the other stabilizers in the formulations, the authors also do not mention the comprehensive study by Allen et. al. on coloration of the closely related SBC. (Polymer Degradation and Stability, 2006, 91:1395.).

Our response: We really appreciate your valuable question. Regarding the comprehensive study by Allen et. al. you mentioned, [Polymer Degradation and Stability, 2006, 91, 1395] the authors compared the difference of similar auxiliary materials in aging performance on K-Resin in detail, and use special functional groups with steric hindrance effects to explain the reason of aforementioned different performance. Besides, the authors discussed two major issues of K-Resin of both thermal aging degradation and photo aging degradation. In addition, comparing the stability of Alkanox P-24, Irgafos 168 and Irganox 1010, the authors also found that the addition of HALS and deactivator can antagonize the stability of polymers containing classic antioxidants. This study is a model for every researcher who investigates the antioxidant process of materials. The difference between our manuscript and this study is that: (1) The comparison of the antioxidants GM, DSPDP, Irganox 1076 and Irganox 3114 mentioned in our manuscript is not covered in the above paper, and the above four antioxidants have definitely wide application prospects. (2) The authors studied the thermal and photo-oxidative stabilization of K-Resin, while our manuscript studied the thermal degradation of Acrylonitrile-Butadiene-Styrene terpolymer under high temperature and high shear conditions. Each monomer structural units of ABS provides different properties to the terpolymer. To summarize, it is necessary to investigate the ABS thermal aging degradation process after the K-Resin thermal aging research. According to your suggestion, to enrich our work, we revised our manuscript.

Our revision:

Line 44: add "In addition, Allen’s group used other stabilizers to conduct the comprehensive study of K-Resin by both thermal aging degradation and photo aging degradation [39]."

Notes and references: add

[26] Allen, N. S.; Barcelona, A.; Edge, M.; Wilkinson, A.; Merchan, C. G.; Quiteria, V. R. S. Aspects of the thermal and photostabilisation of high styrene–butadiene copolymer (SBC). Polymer Degradation and Stability 2006, 91, 1395-1416.

Comment 4

Reviewer wrote: This paper is marginal for publication in any case as the methods used are very routine and there is no real mechanistic contribution to the literature on polymer degradation and stabilization.

Our response: Thanks very much for your suggestions. We apologize for the inconvenience caused by our article when you read them. In our manuscript, we focus on the application of GM to the thermal aging experiment in ABS materials. The key point is to show the difference of stabilizers GM and DSPDP, Irganox 1076 and Irganox 3114 during the aging process, and to change the binary concept of the classic primary antioxidant and secondly antioxidant in additives. The experimental results show that the addition of GM can improve the thermal resistance of ABS in the oven and extruder environment without reducing mechanical properties, which has not been reported in the related literature. Therefore, we think our work is meaningful. We apologize again for the inconvenience caused by our article when you read them.

Round 2

Reviewer 2 Report

The authors have addressed the issues I raised regarding acknowledging the work of others and more appropriately positioning their work in the framework of the literature.

The methods used are routine but the work has been soundly performed; it may be of value to others working with ABS processing.

I am happy to see it go forward for publication.